# A Scoring System for Predicting Microvascular Invasion in Hepatocellular Carcinoma Based on Quantitative Functional MRI

**DOI:** 10.3390/jcm11133789

**Published:** 2022-06-30

**Authors:** Chien-Chang Liao, Yu-Fan Cheng, Chun-Yen Yu, Leung-Chit Leo Tsang, Chao-Long Chen, Hsien-Wen Hsu, Wan-Ching Chang, Wei-Xiong Lim, Yi-Hsuan Chuang, Po-Hsun Huang, Hsin-You Ou

**Affiliations:** 1Department of Radiology, Kaohsiung Chang Gung Memorial Hospital, Chang Gung University College of Medicine, 123 Ta-Pei Road, Niao-Sung District, Kaohsiung 833, Taiwan; liao1009@gmail.com (C.-C.L.); prof.chengyufan@gmail.com (Y.-F.C.); y7192215@ms17.hinet.net (C.-Y.Y.); leolctsang@gmail.com (L.-C.L.T.); lordblue607@yahoo.com.tw (H.-W.H.); o927003551@gmail.com (W.-C.C.); rahxaphon01@gmail.com (W.-X.L.); jenaqwer@gmail.com (Y.-H.C.); qwe79103@gmail.com (P.-H.H.); 2Department of Surgery, Kaohsiung Chang Gung Memorial Hospital, Chang Gung University College of Medicine, 123 Ta-Pei Road, Niao-Sung District, Kaohsiung 833, Taiwan; clchen@adm.cgmh.org.tw

**Keywords:** diffusion-weighted image, microvascular invasion, hepatocellular carcinoma, predictive scoring model

## Abstract

Microvascular invasion (MVI) in hepatocellular carcinoma (HCC) is a histopathological marker and risk factor for HCC recurrence. We integrated diffusion-weighted imaging (DWI) and magnetic resonance (MR) image findings of tumors into a scoring system for predicting MVI. In total, 228 HCC patients with pathologically confirmed MVI who underwent surgical resection or liver transplant between November 2012 and March 2021 were enrolled retrospectively. Patients were divided into a right liver lobe group (*n* = 173, 75.9%) as the model dataset and a left liver lobe group (*n* = 55, 24.1%) as the model validation dataset. Multivariate logistic regression identified two-segment involved tumor (Score: 1; OR: 3.14; 95% CI: 1.22 to 8.06; *p* = 0.017); ADC_min_ ≤ 0.95 × 10^−3^ mm^2^/s (Score: 2; OR: 10.88; 95% CI: 4.61 to 25.68; *p* = 0.000); and largest single tumor diameter ≥ 3 cm (Score: 1; OR: 5.05; 95% CI: 2.25 to 11.30; *p* = 0.000), as predictive factors for the scoring model. Among all patients, sensitivity was 89.66%, specificity 58.04%, positive predictive value 68.87%, and negative predictive value 84.41%. For validation of left lobe group, sensitivity was 80.64%, specificity 70.83%, positive predictive value 78.12%, and negative predictive value 73.91%. The scoring model using ADC_min_, largest tumor diameter, and two-segment involved tumor provides high sensitivity and negative predictive value in MVI prediction for use in routine functional MR.

## 1. Introduction

Recurrence of hepatocellular carcinoma (HCC) after curative treatment such as surgical resection or liver transplantation is the most significant concern when treating HCC. The HCC recurrence rate after surgical resection is 10%, reaching 70–80% in 5 years and affecting 8–20% of patients after liver transplantation [1,2,3].

Microvascular invasion (MVI) in HCC is a histopathological marker and a risk factor for HCC recurrence [4]. MVI is defined as a cancer cell nest in the endothelial vascular lumen under microscopy [5]. Shuji et al. [6] reported that recurrence-free survival rates at two years for patients without MVI, with mild MVI (1–5 invaded vessels), and with severe MVI (more than five invaded vessels) were 75.9, 47.2, and 32.7%, respectively, indicating that severe MVI leads to a high-frequency of recurrence. Predicting the presence of MVI before hepatic resection can help determine surgical strategies. In addition, several biochemical markers are used for predicting MVI, including alpha-fetoprotein (AFP), des-gamma-carboxy prothrombin (DCP), and gamma-glutamyl transferase (GGT) [7,8].

Diffusion-weighted imaging (DWI) is an MR modality for calculating the apparent diffusion coefficient (ADC), a measure of the molecular diffusion magnitude of tissue fluid [9]. DWI has been widely used as a tumor characteristic depiction tool in radiology [10]. Zhao et al. [11] reported that HCCs with MVI had both lower mean and minimum ADC values (ADC_mean_ and ADC_min_) than HCCs without MVI; however, DWI is easily affected by motion [12], especially cardiac motion, which has a marked effect on liver DWI, resulting in higher ADC values of normal liver parenchyma in the left hepatic lobe compared with the right hepatic lobe [13]. ADC values for malignant and benign lesions in the right or left liver lobes are calculated from non-cardiac-gated DW-MRI and are also significantly higher in the left hepatic lobe compared with the right hepatic lobe.

Since MVI is a microscopic finding and preoperative biopsy is not recommended in MVI, diagnosis may be missed due to sampling error, and possible complications such as bleeding and tract seeding may occur [14,15]. Many studies have investigated MVI prediction of image findings, including irregular tumor margin, peri-tumoral enhancement, multiple tumors, and the total tumor volume [16,17,18]. Several scoring systems to predict MVI have been developed by combining these imaging findings and biochemical markers [19,20,21], but none of these scoring systems include tumor ADC values as a predictive factor. This study aimed to integrate DWI and MR image findings of tumors into a clinically accessible scoring system for the prediction of MVI.

## 2. Materials and Methods

### 2.1. Patients

This retrospective study was conducted by enrolling patients from the histopathology database in our institution. The preoperative diagnosis of HCC was made by the typical image appearance of the three-phase enhanced contrast-enhanced magnetic resonance imaging (MRI) including DWI study. A total of 416 patients with complete preoperative MR study were enrolled who underwent hepatectomy or liver transplantation for curative treatment at the Department of General Surgery Kaohsiung Chang-Gung Memorial Hospital from November 2012 to March 2021. The inclusion criteria in this study were: 1. All patients were pathologically confirmed as HCC, focusing on MVI examination. 2. All patients received the complete MRI studies, including DWI. 3. There was no imaging evidence of distant metastasis or liver vessel invasion. 4. The tumors had no necrotic change after radiofrequency ablation (RFA) or trans-catheter arterial chemoembolization (TACE). 5. No pathological evidence of positive resection margin nor MVI existed. 6. Complete laboratory data and pathology report. Exclusions included 16 patients who underwent liver transplantation who had post-RFA or TACE necrotic tumors presented in the specimen; 2 patients with imaging evidence of extra-hepatic seeding tumor; 39 patients lacking comprehensive DWI studies; and 131 patients with suboptimal image quality or failure of DWI study. A total of 228 patients met the inclusion criteria and were enrolled in this study. Using the middle hepatic vein as an anatomic reference to split the liver into right lobe and left lobe, according to the liver lobes of lesions with the lowest mean and minimum ADC value measured or MVI positive lesions, the study patients were divided into a right liver lobe group for constructing the scoring model based on the risk factors identified in multivariate analysis and a left liver lobe group for scoring system validation (Figure 1).

### 2.2. Ethical Considerations

The institutional review board (IRB) of Kaohsiung Chang-Gung Memorial Hospital approved the protocol for this retrospective study of patient medical records. Because of the retrospective design, the IRB waived informed consent from included patients (IRB: 202200169B0).

### 2.3. Collection of Patients’ Demographic and Clinical Data

Patients’ demographic data such as gender and age, and clinical parameters such as underlying hepatitis and Child-Pugh scores, were extracted from medical records. Laboratory studies included results of liver function enzymes (GOT and GPT), total bilirubin, GGT, and serum AFP levels that were obtained closest to the surgery. MR imaging of tumor characteristics, including the number of tumors, the maximum diameter of tumor, liver surface involvement, tumor location, mean ADC value, and minimum ADC value of liver tumor were collected. The average interval in days between MRI and the surgery was 26 days. The pathological findings of MVI were microscopically examined for HCC cells in the endothelial vascular lumen.

### 2.4. MRI Protocols

All MR images were acquired on a 1.5 T scanner (Discovery MR450; GE Healthcare, Milwaukee, WI, USA) and a 12-channel Body Array Coils (1.5T Signa HDxt MR System, GE Healthcare) prepared. The MR protocols included the following: breath-hold in-phase and opposed-phase fast spoiled gradient-echo T1 weighted imaging; breath-hold fast spin-echo (FSE) T2-weighted imaging; single-shot FSE heavily T2-weighted imaging with fat suppression; and DWI using SE single-shot echo-planar technique with b-values of 0 and 400 s/mm^2^ (repetition time/echo time (TR/TE): 2400/44; slice thickness/gap: 5/1 mm; matrix: 80 × 128). ADC maps were calculated automatically. Dynamic images using fat-suppressed T1-weighted gradient-echo images with a three-dimensional (3D) acquisition sequence (liver acquisition with volume acceleration, LAVA) were obtained at 30 s, 60 s, and 3 min after IV administration of gadolinium-based contrast medium (Magnevist^®^, BAYER Radiology, Pittsburgh, PA, USA) injection (0.1 mmol/kg, total 10–15 mL). The images were acquired in the transverse plane and had a section thickness of 5 mm (TR/TE: 3.4/1.6; flip angle: 12°; matrix: 288 × 192). MR cholangiography and MR angiography were performed using Coronal 3D MRCP (TR/TE: 3000/505 ms, flip angle: 90°, matrix size: 320 × 320) and breath-hold 3D spoiled gradient-echo imaging (TR/TE, 3.6/1.42, flip angle: 30°; field of view, 40 × 36 cm; bandwidth, 32 kHz; matrix size, 288 × 192).

### 2.5. Image Analysis

Two staff members (O H.-Y. and Liao C.-C.), who have specialized in gastrointestinal and hepatobiliary imaging for 15 years and 6 years, respectively, interpreted the preoperative liver MR studies to record tumor characteristics, including the tumor number, the largest single tumor diameter, summation of multiple tumor diameters, and tumor involving more than two Couinaud segments of the liver. The largest single tumor diameter was defined as the maximum diameter in the transverse plane images and was measured in the portal venous phase or T2-weighted images with clearer tumor boundaries. Quantitative measurements were performed on the DWI and ADC maps using the picture and archiving communication system (GE Healthcare, Barrington, IL, USA). DWI was viewed and compared with referenced T2-weighted and contrast-enhanced T1-weighted transverse images for tumor detection and anatomic correlation. The regions of interest (ROI) of mean and minimum ADC values of the tumor (ADC_mean_ and ADC_min__)_ were manually placed at the cross-section of each viable tumor’s maximum diameter, including the whole lesion on a single slice.

### 2.6. Statistical Analysis

An independent *t*-test was used for continuous variables and a chi-square test for categorical variables. Inter-observer agreement was assessed by the intra-class correlation coefficient (ICC) for the ADC value measurements. ICC values were categorized into 5 categories: 0.0 to 0.20 as poor; 0.21 to 0.40 as fair; 0.41 to 0.60 as moderate; 0.61 to 0.80 as good; and 0.81 to 1.00 as excellent. Predictive factors of continuous variables for MVI with a statistically significant difference (*p* < 0.05) were collected for plotting the receiver operating characteristic (ROC) curves and calculating the area under the ROC curve (AUROC). The predictive factors with AUROC ≥ 0.7 were entered for the cut-off value calculation. All statistically significant predictive factors for continuous variables were categorized into two groups below or above the cut-off value.

All categorized predictive factors were entered into a multivariate logistic regression to calculate the odds ratios (OR) and confidence intervals (CI), and variables with statistically significant differences (*p* < 0.05) were chosen for the scoring system. The scoring system was based on the partial regression coefficient (β) modified from the method of Sullivan et al. [22] and Zhao et al. [19] by designating the predictive factor with the lowest β value as the constant with point 1. The ratio of other predictive factors when divided by the constant was rounded to the nearest integer as the corresponding point. The overall predictive score of an individual patient was calculated by adding the points for all the relevant predictive factors. Both all studied patients and the left lobe group were validated by using the proposed model. The ability of the overall score cut-off value to predict MVI was evaluated using ROC curve analysis. Statistical analysis was performed using commercially available software (SPSS version 22.0, IBM SPSS, Chicago, IL, USA).

## 3. Results

### 3.1. Patients’ Characteristics

Data of 228 patients were enrolled as the analytic sample, including 153 males (mean age: 59.74 ± 10.50 years) and 75 females (mean age: 63.35 ± 9.30 years). Among these, 172 patients had a single HCC tumor, and 56 had multiple HCC tumors; 61 patients received liver transplantation and 167 patients received curative hepatectomy. All resected specimens underwent microscopic examination for MVI; 112 patients had negative MVI and 116 patients had positive MVI.

Patients were divided into a right liver lobe group (*n* = 173, 75.9%) as the model dataset and a left liver lobe group (*n* = 55, 24.1%) as the model validation dataset (Table 1). No significant differences were found between the right liver lobe group and the left liver group in preoperative baseline characteristics, including age (60.38 ± 10.41 years and 62.67 ± 9.58 years, *p* = 0.151), gender (*p* = 0.719), hepatitis (*p* = 0.304), ADC_min_ of liver tumor (0.85 ± 0.36 × 10^−3^ mm^2^/s and 0.96 ± 0.45 × 10^−3^ mm^2^/s, *p* = 0.101), AFP (50.95 ± 122.13 ng/mL and 98.54 ± 177.87 ng/mL, *p* = 0.077), GGT (37.99 ± 38.99 U/L and 31.81 ± 29.95 U/L, *p* = 0.282), GOT (45.78 ± 38.53 U/L and 41.36 ± 23.29 U/L, *p* = 0.675), GPT (40.17 ± 29.39 U/L and 45.80 ± 62.11 U/L, *p* = 0.361), total bilirubin (1.38 ± 1.71 mg/dL and 9.79 ± 52.27 mg/dL, *p* = 0.281), and postoperative incidence of MVI (85 patients, 49.1%, and 31 patients, 56.4%, *p* = 0.350). However, the left liver lobe group had a significantly higher ADC_mean_ value, indicating a statistically significant difference between the left and right liver lobe groups (1.56 ± 0.53 × 10^−3^ mm^2^/s in the left liver lobe group and 1.39 ± 0.32 × 10^−3^ mm^2^/s in the right liver lobe group, *p* = 0.026). In the right liver lobe group, which was used for scoring model development, 88 patients had negative MVI and 85 patients had positive MVI; in the left liver lobe group, 24 patients had negative MVI and 31 patients had positive MVI.

### 3.2. Univariate Factors for MVI Prediction Based on the Model Dataset of the Right Liver Lobe Group

In the right liver lobe group, no significant differences were found between MVI negative and positive patients in preoperative characteristics (Table 2), including age (59.98 ± 8.90 years and 60.80 ± 11.80 years, *p* = 0.607), gender (*p* = 0.873), hepatitis (*p* = 0.769), AFP (36.67 ± 77.77 ng/mL and 67.52 ± 157.80 ng/mL, *p* = 0.127), GGT (41.03 ± 41.38 U/L and 34.85 ± 36.31 U/L, *p* = 0.298), GOT (45.83 ± 31.92 U/L and 45.72 ± 44.53 U/L, *p* = 0.985), GPT (39.33 ± 30.58 U/L and 41.04 ± 28.27 U/L, *p* = 0.704), total bilirubin (1.54 ± 1.84 mg/dL and 1.21 ± 1.55 mg/dL, *p* = 0.208).

Among the significant univariate factors entered the multivariate analysis of MVI prediction of 173 patients from the right liver lobe group, significant differences were found in the largest tumor diameter, ADC values of liver tumor, and tumors involving more than two segments of the liver (*p* < 0.05) compared to other factors. The largest tumor diameter was higher in the MVI positive group than in the MVI negative group (5.24 ± 3.32 cm and 2.70 ± 1.61 cm, *p* < 0.05). The ADC_mean_ and ADC_min_ of the positive MVI group were lower than those in the negative group (1.26 ± 0.27 mm^2^/s and 1.52 ± 0.31 × 10^−3^ mm^2^/s; 0.68 ± 0.28 × 10^−3^ mm^2^/s and 1.02 ± 0.34 × 10^−3^ mm^2^/s, *p* < 0.05). The ICC of both readers were good in ADC_mean_ (0.70, 95% CI: 0.59 to 0.78) and excellent in ADC_min_ (0.82, 95% CI: 0.74 to 0.87). 35 patients with MVI positive lesions had more than two segments of the liver involved, and 11 patients with MVI negative lesions had more than two segments of the liver involved (*p* < 0.05).

### 3.3. ROC Curve of Predictive Factors for the Cut-Off Value

Three continuous variables among predictive factors reached statistical differences for ROC curve plotting between MVI positive and negative groups (Figure 2), including the largest tumor diameter (AUROC: 0.76, *p* = 0.000), ADC_mean_ (AUROC:0.74, *p* = 0.000), and the ADC_min_ (AUROC:0.79, *p* = 0.000). Those factors with AUROC ≥ 0.7 were entered for the cut-off value calculation. The cut-off value was 3 cm of the largest single tumor, 1.38 × 10^−3^ mm^2^/s of ADC_mean_, and 0.95 × 10^−3^ mm^2^/s of ADC_min_.

### 3.4. Multivariate Logistic Regression for Predictive Factors and Scoring Models

Four categorized predictive factors were enrolled for multivariate logistic regression. Variables with a statistically significant difference (*p* < 0.05) in the univariate regression analysis were chosen for the scoring system, including ADC_mean_, ADC_min,_ the largest single tumor diameter, and the two-segment involved tumor. Two scoring models (model A and model B) were built to evaluate their effects on predicting MVI. A corresponding score was designated for all factors to establish the scoring model. The predictive factors and each score of model A consisted of a two-segment involved tumor (Score: 1; OR: 3.41; 95% CI: 1.27 to 9.16; *p* = 0.015), ADC_mean_ ≤ 1.38 × 10^−3^ mm^2^/s (Score: 1; OR: 5.45; 95% CI: 2.15 to 13.81; *p* = 0.000), ADC_min_ ≤ 0.95 × 10^−3^ mm^2^/s (Score: 2; OR: 6.96; 95% CI: 2.79 to 17.37; *p* = 0.000), the largest single tumor diameter ≥ 3 cm (Score: 2; OR: 7.90; 95% CI: 3.11 to 20.04; *p* = 0.000). The predictive factors and each score of model B consisted of a two-segment involved tumor (Score: 1; OR: 3.14; 95% CI: 1.22 to 8.06; *p* = 0.017), ADC_min_ ≤ 0.95 × 10^−3^ mm^2^/s (Score: 2; OR: 10.88; 95% CI: 4.61 to 25.68; *p* = 0.000), the largest single tumor diameter ≥ 3 cm (Score: 1; OR: 5.05; 95% CI: 2.25 to 11.30; *p* = 0.000). (Table 3) The predictive ability of the scoring models was evaluated with the ROC curve (Figure 3). The AUROC of model A was 0.84 (95% CI: 0.79 to 0.89; *p* = 0.000). The AUROC of model B was 0.82 (95% CI: 0.76 to 0.87; *p* = 0.000). The optimal cut-off score was 3 in model A and 2 in model B.

### 3.5. Validation of the Proposed Scoring Models

The summation of predictive factor scores determined the probability of MVI in all patients in the total study group (Table 4). The probability of MVI of model A is in the proportion of the total score with 7.32% (3/41) in score 0, 25% (3/12) in score 1, 22.86% (8/35) in score 2, 59.68% (37/62) in score 3, 65.38% (17/26) in score 4, 91.18% (31/34) in score 5, and 94.44% (17/18) in score 6. The probability of MVI of model B is in the proportion of the total score with 11.76% (6/51) in score 0, 23.08% (6/26) in score 1, 54.05% (40/74) in score 2, 78.00% (39/50) in score 3, and 92.59% (25/27) in score 4. In the validation for model A in the left lobe group, the probability of MVI of model A is in the proportion of the total score with 13.33% (2/15) in score 0, 100.00% (1/1) in score 1, 50.00% (6/12) in score 2, 76.92% (10/13) in score 3, 71.43% (5/7) in score 4, and 100% (7/7) in score 5. In the validation of model B of the left lobe group, the probability of MVI of model B is in the proportion of the total score with 18.75% (3/16) in score 0, 42.86% (3/7) in score 1, 72.22% (13/18) in score 2, and 85.71% (12/14) in score 3. In the overall patient validation, model A had a sensitivity of 87.93%, specificity 66.07%, and positive predictive value was 72.86% and a negative predictive value was 84.09%; model B had a sensitivity of 89.66%, specificity 58.04%, and positive predictive value was 68.87% and negative predictive value was 84.41%. In the validation of patients in the left lobe group, model A had a sensitivity of 70.97%, specificity 79.17%, and positive predictive value was 81.48% and a negative predictive value was 67.86%; model B had a sensitivity of 80.64%, specificity 70.83%, and positive predictive value was 78.12% and negative predictive value was 73.91%.

## 4. Discussion

Results of the present study show that quantitative functional MRI predicts MVI in HCC using a simple quantitative way to avoid equivocal imaging findings or inter-reader and intra-reader bias. In the present study, two of the proposed scoring models comprised four predictive factors, including ADC_mean_ ≤ 1.38 × 10^−3^ mm^2^/s, ADC_min_ ≤ 0.95 × 10^−3^ mm^2^/s, the largest single tumor diameter ≥ 3 cm, and tumor involving more than two Couinaud segments of the liver. The designated score for each integrated predictive factor is weighted on the partial regression coefficient. A calculated score of more than 3 points in model A and 2 points in model B indicates a high risk of MVI with high sensitivity and negative predictive value. The ROC curve of the scoring models showed an AUROC of 0.84 and 0.82, and these values are higher than a single predictive factor, thus combining these factors predicts MVI more accurately.

In 2013, Rodríguez-Perálvarez et al. [23] conducted a systematic review of MVI and concluded that MVI is a marker of aggressive HCC behavior that changes the disease prognosis after curative therapy. A more recent systematic review by Erstad and Tanabe in 2019 still concluded that MVI is a critical determinant of recurrence after surgical resection and liver transplantation, and although MVI has not been incorporated in common HCC staging systems [24], it may be used to help surgeons determine surgical interventions [25].

DWI is useful in liver lesion detection and characterization based on the differential signal intensities between the lesion and liver parenchyma. Tissues with high cellularity usually demonstrate limited diffusion, which shows high signal intensity at higher b values and a low ADC value [9]. The differential diagnosis for many liver lesions requires recognition of specified contrast medium enhancement patterns either in CT or MRI. However, the DWI study does not need intravenous contrast medium injection, which provides additional tissue information and applies to patients with impaired renal function [26]. Suh et al. [27] proposed a possible mechanism for the relationship between DWI and MVI in 2012, indicating that DWI is associated with the histological grade of HCC, and the ADC values are higher in well or moderately differentiated HCCs than in poorly differentiated HCCs [28,29]. This result can be explained by the increased nucleus/cytoplasm ratio in poorly differentiated HCCs, which would theoretically decrease the free diffusion of water molecules in the intracellular space causing a lower ADC value [30]. The MVI of HCCs has similar results for ADC values as in histological grade HCCs based on results of meta-analysis [31]. First, however, the direct evidence of whether HCCs with MVI have similar high cellularity as that in poorly differentiated HCCs is not clear; and second, the influence of decreased micro-capillary perfusion in HCCs with MVI may also cause ADC values to decrease [11,32].

A significantly higher ADC_mean_ value of liver tumors was noticed in the left liver group with statistically significant differences between the left and right liver lobe groups. This observation was compatible with those in previous studies [12,13]; however, we found that there was no significant difference in ADC_min_ value in HCC of both liver lobes, indicating that ADC_min_ is a relatively stable value that is less affected by cardiac motion. Given that ADC measurement in the left liver lobe lesion may be incorrect in a non-cardiac gated MRI study [33], the measured ADC values by which to construct the scoring models were from patients with HCCs in the right liver lobe to avoid discrepancy of ADC_mean_ in left lobe HCC, and the left liver lobe group was used as an external validation test.

In the present study, the main difference between the two proposed models was the lack of ADC_mean_ in model B. ADC_mean_ was mostly used in previous studies to discuss the differentiation of HCC histological grade, which have different results, including three studies [28,29,34] that found highly differentiated HCCs had higher ADC_mean_ values than moderate and poorly differentiated tumors, and two studies [35,36] that reported the histopathologic grade of HCC did not correlate with the ADC_mean_. Similar controversial results regarding ADC_mean_ in the prediction of HCC with MVI were reported by two studies stating that ADC_mean_ is significantly lower in HCCs with MVI than in HCCs without MVI [37,38]. However, another three studies reported no significant differences between HCCs with MVI and HCCs without MVI [39,40,41]. To date, no standardized ADC measuring method has been available for liver DWI study. Some researchers tend to draw an ROI avoiding the cystic necrotic part, and the hemorrhage of the measured lesion [28,29,35,38,40,42,43], while some draw the ROI on the entire liver tumor [27,34,36,37,39,41]. This may explain the diversity of ADC_mean_ measurements and the correlation between MVI and the histo-pathologic grade of HCC. It usually requires other MRI pulse sequences such as enhanced images, T2 and T1 weighted images to identify the cystic necrotic part. In the present study, we drew the ROI on the entire liver tumor because it seems to be an easier way to obtain the ADC_mean_ without spending more time determining the unwanted part of the measurement, which also makes it more clinically useful in daily practice.

A meta-analysis by Surov et al. [31] in 2020 focused on the role of ADC values in the prediction of MVI in HCC, specifically by measuring the mean and minimum ADC values of the tumor (ADC_mean_ and ADC_min_). These authors concluded that ADC_min_ was able to predict MVI. Theoretically, it was suggested that regions of minimum ADC reflected the highest cellular zone [44]. In the present study, both ADC_mean_ and ADC_min_ reached a statistically significant lower value in HCC with MVI; however, ICC results were only good in ADC_mean_ but were excellent in ADC_min_, (0.70 vs. 0.82), suggesting that some level of inter-observer disagreement was present in ADC_mean_, possibly because when drawing ROI in the entire lesion, it is inevitable to include both the cystic necrotic part and the highest cellular zone of the tumor. Therefore, the ADC_mean_ may vary due to the heterogeneous part of the tumor, while the ADC_min_ value of a tumor corresponds with the highest tumor cellularity and, in the measurement of ADC values, may be less variable and more reproducible. In the present study, when ADC_mean_ was removed from model A to build model B, the logistic regression of model B with the other three predictive factors revealed the highest odds ratio of ADC_min_ (OR: 10.88; 95% CI: 4.61 to 25.68; *p* = 0.000). The results of model B validation in the validation dataset (left lobe group) showed better sensitivity (80.64% vs. 70.97%) and negative predictive value (73.91% vs. 69.85%), therefore, model B is our preferred scoring system for predicting MVI.

The definition of the liver segment described by Couinaud is an independently functional unit receiving a hepatic artery, a portal vein, and drained by a hepatic vein [45]. The location of HCC is currently considered to be an important factor influencing both diagnostic strategy and therapeutic options [46]. In 2019, Al-Azzawi et al. [47] reported that the risk of MVI in segment 8 HCC was 3.5 times higher than the risk from the other segments. HCCs located in segments 4, 5, and 8 are defined as central HCCs [48]. The characteristic of the central HCCs is receiving a dual blood supply from both left and right hepatic arteries, which carries an increased risk of MVI [49]. We assumed that HCC involving two segments theoretically has a higher chance of multiple vascular invasions, thus increasing the risk of MVI for a similar reason as with central HCCs. However, studies addressing whether HCC with MVI correlates with involving two liver segments are scarce. In the present study, tumors involving more than two segments of the liver meet the statistically significant difference (*p* < 0.05) for associations with MVI in the model dataset (right liver lobe group), therefore, we included this factor in our scoring model.

A series of studies were conducted to predict MVI preoperatively using serum markers such as AFP, the agglutinin-reactive fraction of AFP (AFP-L3), GGT, des-gamma-carboxy prothrombin (DCP), and liver function enzymes [20,50,51]. However, besides DCP, which was not available in the routine pre-operative collection of our institution, no significant differences were found in AFP, GOT, GPT, total bilirubin, and GGT between MVI positive and negative patients in the model dataset (right liver lobe group). This may be because these serum markers were collected at various intervals of time before surgery, which may result in bias.

The liver imaging reporting and data system (LI-RADS) which was introduced by the American College of Radiology is a system of standardized terminology and a classification system for imaging findings in liver lesions. Its score classifies a liver lesion’s relative risk for HCC from LR-1 (definitely benign) to LR-5 (definitely HCC). [52] Based on the latest version of Li-RADS revised in 2018 (LI-RADS v2018), Chen et al. in 2019 [53] concluded that the categorized LR-5 lesion with mosaic architecture and non-smooth tumor margin independently predicted MVI, and Centonze et al. in 2022 [54] reported the LR-5 lesion associated with unfavorable pathological characteristics including MVI (41.7%), satellitosis (25.8), and capsule infiltration (28%), in LR 3/4 lesion, there was a non-negligible percentage of MVI (22.6%), satellitosis (9.4%), and capsule infiltration (11.3%). These results suggested that the higher LI-RADS category may predict MVI but the intermediate-risk LI-RADS category is not warranted as MVI negative. Previous studies [38,55] reported several MR image parameters to predict MVI such as tumor capsule, tumor margin, peri-tumoral enhancement, and peri-tumoral hypo-intensity in the hepatobiliary phase after gadoxetic acid contrast medium usage. Some of these image findings are included in the ancillary features of LI-RADS v2018. Few studies [56,57] have discussed the ADC value in the characterization of hepatic focal lesions in correlation to LI-RADS. In future work, integration of the quantitative MR study with the ancillary feature of LI-RADS might be helpful in improving MVI prediction between the high and intermediate-risk categories of LI-RADS.

This study has several limitations. First, although we did our best to enroll consecutive patients, there were many patients lacking preoperative MR studies during the early years of the studied interval, or suboptimal image quality and failure of DWI resulted in a limited number of studied patients as the major limitation of this research; besides, the nature of the retrospective study inevitably has selection bias. Second, the method of ADC value measurement by drawing ROI in the entire lesion differs from several previous studies, which may result in measurement bias, even though inter-observer agreement was reached. Still, further consensus on the measurement of ADC values is required to reach more reproducible results. Third, a 1.5 test MR scanner and only two b-values of 0 and 400 s/mm^2^ were conducted in this study, which may result in measurement errors compared to using multiple b values; however, using more b values extends the scan time of the MR study. Finally, no relevant data are available regarding our scoring model in the validation of recurrent HCC of studied patients, which requires prospective study to further verify the effectiveness of this predictive model.

## 5. Conclusions

In conclusion, the scoring model comprised of ADC_min_, the largest tumor size, and the tumor involving more than two Couinaud segments of the liver provides high sensitivity and negative predictive value and thus can be used as the initial predictive factor for MVI in patients with HCC. Besides using the model in quantitative MR studies, given that all parameters in this model do not necessarily require intravenous contrast medium usage, it can also be applied in patients with renal function impairment.

## Figures and Tables

**Figure 1 jcm-11-03789-f001:**
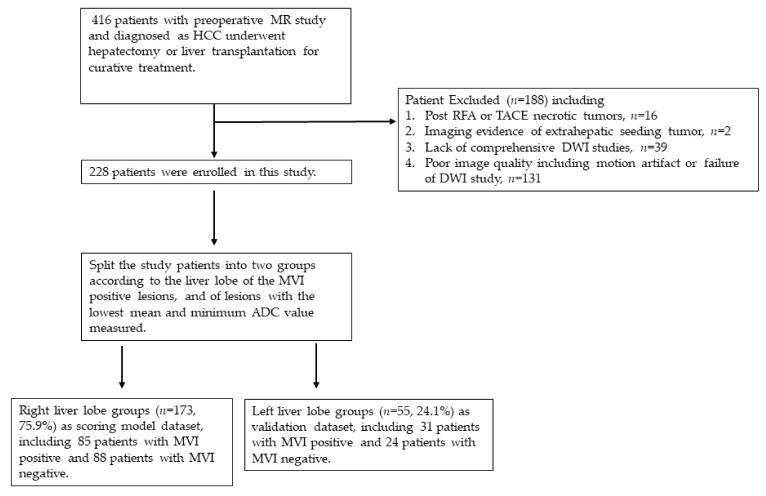
Flow chart of patient enrollment.

**Figure 2 jcm-11-03789-f002:**
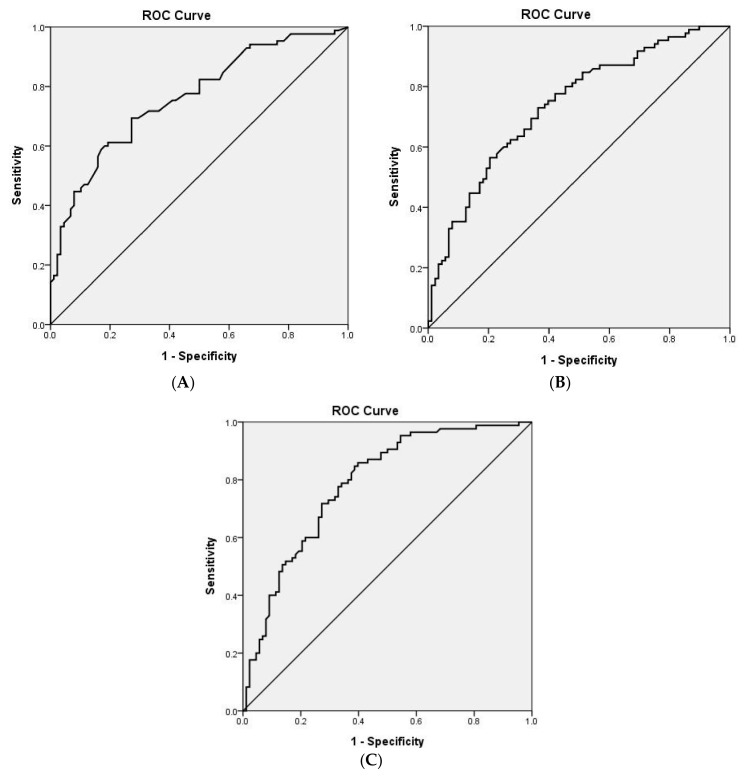
ROC curve analysis of the predictive factor for MVI in HCC. The area under curve of the largest tumor diameter (**A**), ADC_mean_ (×10^−3^ mm^2^/s) (**B**), ADC_min_ (×10^−3^ mm^2^/s) (**C**), were 0.76, 0.74, and 0.79 respectively, *p* value < 0.05. The cut-off value was 3 cm for the largest single tumor, 1.38 × 10^−3^ mm^2^/s of ADC_mean_ and 0.95 × 10^−3^ mm^2^/s of ADC_min_.

**Figure 3 jcm-11-03789-f003:**
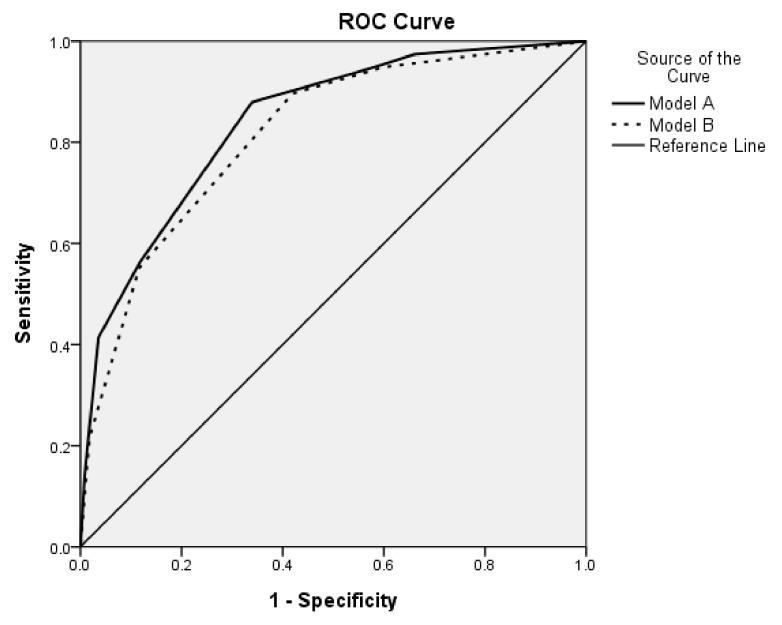
The ROC curve of two proposed scoring models. In model A, AUROC was 0.84 (95% CI: 0.79–0.89) and the *p* value was 0.000; in model B AUROC was 0.82 (95% CI: 0.76–0.87) and the *p* value was 0.000; the cut-off value of the score in model A was 3 points and 2 points in model B.

**Table 1 jcm-11-03789-t001:** Demographic and clinical characteristics of patients are divided into the model dataset (right liver lobe group) and validation dataset (left liver lobe group).

Characteristics	Total (*n* = 228)	Model Dataset (*n* = 173)	Validation Dataset (*n* = 55)	*p* Value
Age (years)	60.93 ± 0.68	60.38 ± 10.41	62.67 ± 9.58	0.151
Gender	-	-	-	0.719
Male	153	115 (66.5%)	38 (69.1%)	-
Female	75	58 (33.5%)	17 (30.9%)	-
Hepatitis	-	-	-	0.304
HBV	119	90 (52%)	29 (52.7%)	-
HCV	92	67 (38.7%)	25 (45.5%)	-
HBV+HCV	3	3(1.7%)	0 (0.0%)	-
Non-B and non-C hepatitis	14	13 (7.5%)	1 (1.8%)	-
AFP (ng/mL)	62.51 ± 9.49	50.95 ± 122.13	98.54 ± 177.87	0.077
GGT (U/L)	36.50 ± 2.45	37.99 ± 38.99	31.81 ± 29.95	0.282
GOT (U/L)	44.71 ± 2.35	45.78 ± 38.53	41.36 ± 23.29	0.675
GPT (U/L)	41.53 ± 2.63	40.17 ± 29.39	45.80 ± 62.11	0.361
Total bilirubin (mg/dL)	3.41 ± 1.87	1.38 ± 1.71	9.79 ± 52.27	0.281
Diameter (cm)	-	-	-	-
Summation of tumor diameter	5.63 ± 14.87	6.16 ± 16.97	3.93 ± 2.92	0.335
Single tumor diameter	3.77 ± 2.71	3.94 ± 2.89	3.19 ± 1.99	0.074
Tumor number	-	-	-	0.370
Single	172	133 (76.9%)	39 (70.9%)	-
Multiple	56	40 (23.1%)	16 (29.1%)	-
Single tumor involving more than two-segment	48	46 (26.6%)	2 (3.6%)	0.000
ADC_mean_ (×10^−3^ mm^2^/s)	1.43 ± 0.26	1.39 ± 0.32	1.56 ± 0.53	0.026
ADC_min_ (×10^−3^ mm^2^/s)	0.88 ± 0.25	0.85 ± 0.36	0.96 ± 0.45	0.101
MVI	116	85 (49.1%)	31 (56.4%)	0.350

**Table 2 jcm-11-03789-t002:** The predictive factors for the MVI of HCCs based on right lobe lesion.

Factors	MVI Negative (*n* = 88)	MVI Positive (*n* = 85)	*p* Value
Age (years)	59.98 ± 8.90	60.80 ± 11.80	0.607
Gender	-	-	0.873
Male	58 (65.9%)	57 (67.1%)	-
Female	30 (34.1%)	28 (32.9%)	-
Hepatitis	-	-	0.769
HBV	44 (50.0%)	46 (54.1%)	-
HCV	37 (42.0%)	30 (35.3%)	-
Non-B and non-C hepatitis	6 (6.8%)	7 (8.2%)	-
HBV+HCV	1 (1.1%)	2 (2.4%)	-
AFP (ng/mL)	36.67 ± 77.77	67.52 ± 157.80	0.127
GGT (U/L)	41.03 ± 41.38	34.85 ± 36.31	0.298
GOT (U/L)	45.83 ± 31.92	45.72 ± 44.53	0.985
GPT (U/L)	39.33 ± 30.58	41.04 ± 28.27	0.704
Total bilirubin (mg/dL)	1.54 ± 1.84	1.21 ± 1.55	0.208
Diameter (cm)	-	-	-
Summation of tumor diameter	6.16 ± 23.43	6.17 ± 4.55	0.294
Largest tumor diameter	2.70 ± 1.61	5.24 ± 3.32	0.000
Tumor number	-	-	0.188
Single	64 (72.7%)	69(81.2%)	-
Multiple	24 (27.3%)	16(18.8%)	-
Single tumor involving more than two-segment		--	0.000
No	77 (87.5%)	50 (58.8%)	-
Yes	11 (12.5%)	35 (41.2%)	-
ADC value measurement	-	-	-
ADC_mean_ (×10^−3^ mm^2^/s)	1.52 ± 0.31	1.26 ± 0.27	0.000
ADC_min_ (×10^−3^ mm^2^/s)	1.02 ± 0.34	0.68 ± 0.28	0.000

**Table 3 jcm-11-03789-t003:** Multivariate logistic regression analysis of two proposed scoring models.

	β	*p* Value	OR	95% CI for OR		Score
**Predictive Factors of Model A**				Lower	Upper	
Single tumor involving more than two-segment	1.226	0.015	3.406	1.266	9.162	1
ADC_mean_ ≤ 1.38 × 10^−3^ mm^2^/s	1.695	0.000	5.449	2.150	13.807	1
Single tumor ≥ 3 cm	2.067	0.000	7.900	3.114	20.037	2
ADC_min_ ≤ 0.95 × 10^−3^ mm^2^/s	1.940	0.000	6.961	2.789	17.371	2
**Predictive Factors of Model B**						
Single tumor involving more than two-segment	1.144	0.017	3.139	1.222	8.060	1
Single tumor ≥ 3 cm	1.618	0.000	5.046	2.253	11.299	1
ADC_min_ ≤ 0.95 × 10^−3^ mm^2^/s	2.387	0.000	10.882	4.611	25.681	2

β = partial regression coefficient; OR = odds ratio; CI = Confidence Interval.

**Table 4 jcm-11-03789-t004:** Performance of two scoring models.

Total Patients	MVI-Positive Patients (*n* = 116)	Patient Number (*n* = 228)	Probability of MVI	
Model A Score				*p* value = 0.000
0	3	41	7.32%	
1	3	12	25%	
2	8	35	22.86%	
3	37	62	59.68%	
4	17	26	65.38%	
5	31	34	91.18%	
6	17	18	94.44%	
Model B score				*p* value = 0.000
0	6	51	11.76%	
1	6	26	23.08%	
2	40	74	54.05%	
3	39	50	78.00%	
4	25	27	92.59%	
Validation group	MVI-positive patients (*n* = 31)	Patient number (*n* = 55)	Probability of MVI	
Model A score				*p* value = 0.000
0	2	15	13.33%	
1	1	1	100%	
2	6	12	50%	
3	10	13	76.92%	
4	5	7	71.43%	
5	7	7	100%	
Model B score				*p* value = 0.000
0	3	16	18.75%	
1	3	7	42.86%	
2	13	18	72.22%	
3	12	14	85.71%	

## Data Availability

The data that support the findings of this study are not openly available due to human data and are available from the corresponding author upon reasonable request.

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
