# Peer review of "A Scoring System for Predicting Microvascular Invasion in Hepatocellular Carcinoma Based on Quantitative Functional MRI"

_jcm, 2022, doi:10.3390/jcm11133789_

Round 1

Reviewer 1 Report

1. This article mentioned that the ADC mean values of tumors in the left and right lobes of the liver were different, so why were the patients divided into development and validation sets according to the lesions location in the left and right lobes of the liver ?

2. Since there were models for MRI to predict MVI before, what advantages and differences did this model have.

Author Response

Dear Editors and reviewers of JCM

  Thanks for your review of our manuscript (ID: jcm-1783368) entitled A scoring system for predicting microvascular invasion in hepatocellular carcinoma based on quantitative functional MRI, which has been recently returned along with the reviewer’s comments and suggestions.

Comments and Suggestions for Authors

Point 1. This article mentioned that the ADC mean values of tumors in the left and right lobes of the liver were different, so why were the patients divided into development and validation sets according to the lesions’ location in the left and right lobes of the liver?

Author Response to point 1:

  Thanks for your comment. After reviewing several studies regarding the difference in ADC values of liver tumors and parenchyma between right and left liver lobes, they found higher ADC values of either liver tumor or parenchyma of the left lobe liver than that in the right lobe liver, this is possible due to the signal loss of DWI during heart beating since the left lobe of the liver is close to the inferior heart border resulted in the increment of ADC value. In our study, we had a similar observation of ADC mean values which are higher in the left lobe HCC (1.56±0.53) versus right lobe HCC (1.39±0.32). However, most of these studies measured ADC mean values instead of measuring both ADC mean and minimum values, we found that there was no significant difference of ADC minimum value in the right or left lobe HCC in our study, indicating that ADC minimum is a relatively stable value which is less affected by the cardiac motion, this is possible due to the ADC minimum reflect the highest cellularity of the tumor.

  The reason we choose to use right lobe HCC as our model dataset is to avoid such discrepancy of ADC mean in left lobe HCC, and we want to make sure that the scoring model built from right lobe HCC can be used in HCC in both lobes that increase the clinical usefulness.

Author action to point 1:

We will re-edit the paragraph in the discussion section to address this finding.

Page 11, paragraph 4 of discussion, line 307-315

“A significantly higher ADCmean value of liver tumors were noticed in the left liver group with statistically significant differences between the left and right liver lobe groups. This observation was compatible with those in previous studies [24,25], however, we found that there was no significant difference in ADCmin value in HCC of both liver lobes, indicating that ADCmin is a relatively stable value that is less affected by cardiac motion. Given that ADC measurement in the left liver lobe lesion may be incorrect in a non-cardiac gated MRI study [26], the measured ADC values by which to construct the scoring models were from patients with HCCs in the right liver lobe to avoid discrepancy of ADCmean in left lobe HCC, and the left liver lobe group was used as an external validation test.”

  1. Since there were models for MRI to predict MVI before, what advantages and differences did this model have.

Author Response to point 2:

  Thanks for your comment, previous studies have reported several image parameters to predict MVI such as tumor capsule, tumor margin, peritumoral enhancement, and peritumoral hypointensity in the hepatobiliary phase after gadoxetic acid contrast medium usage. Our proposed model differs from these studies with all parameters were collected without contrast medium injection which offers an option for renal function impairment patients in preoperative survey. Besides, while ADC mean and minimum values measurement was conducted in several previous studies and demonstrated the potential usefulness of ADC minimum value in predicting MVI, in this study we reported integration of ADC minimum value, tumor size, and tumor involved segment as a rapid and simple quantitative way to avoid equivocal imaging finding or interreader and intrareader bias.

Author action to point 2:

We will address this issue along with the application of LI-RADS suggested by the other reviewer in the discussion section.

We have studied your comments carefully and have made amendments which are marked in red in the manuscript. We have tried our best to revise our manuscript in accordance with the comments, and recheck the English grammar in the article by another colleague.

We would like to express our gratitude to you for the comments and suggestions regarding our manuscript. It will be highly appreciated if the paper in its present revised form is accepted for publication in your esteemed journal. If there are any further comments or necessary adjustments, please do not hesitate to let us know.

Thank you and best regards.

Dr. Chien Chang Liao
Department of Radiology
Kaohsiung Chang-Gung Memorial Hospital, Taiwan.
+886 975 056 724
liao1009@gmail.com

Reviewer 2 Report

This original paper from Liao and coll. entitled "A scoring system for predicting microvascular invasion in hepatocellular carcinoma based on quantitative functional MRI" analyse application of MVI for prediction of MVI in HCC setting

The clinical impact of MVI in HCC setting is well recognized, as prediction of MVI could provide useful information for potentially switch the management of the HCC patient (i.e. neoadjuvant treatments? transplantation?)

The study is well designed, although it has a small sample size that represent the strongest limitation

I would like to ask the Authors to address some points:

1) please expand the flowchart for patient selection presented in Picture 1 in order to provide an idea of the overall volume of the center (285 patients over a 9 year seem quite low, but probably there were a lot of patients that were excluded)

2) discuss the role of LI-RADS classification as a potential and easily applicable predictor of pathological features of HCC (10.3390/diagnostics12010160)

3) Include the limited number of patients as a major limitation of the study

Congratulations for this interesting research

Best regards.

Author Response

Dear Editors and reviewers of JCM

  Thanks for your review of our manuscript (ID: jcm-1783368) entitled A scoring system for predicting microvascular invasion in hepatocellular carcinoma based on quantitative functional MRI, which has been recently returned along with the reviewer’s comments and suggestions.

Point 1: please expand the flowchart for patient selection presented in Picture 1 in order to provide an idea of the overall volume of the center (285 patients over a 9 year seem quite low, but probably there were a lot of patients that were excluded)

Author Response to point 1:

  Thanks for your comment and suggestion, initially we enrolled our case from the pathology report database in our institution, however, in the early year from 2012 to 2014, preoperative MR was not routinely performed, and most patients received surgery based on classic CT findings (cirrhotic liver with the typical enhanced pattern). Although we mentioned our enrolled time interval was from 2012 in the method section, most (161 patients out of 228 patients) of the enrolled cases were collected from 2017 to 2021, from 2012 to 2016, although several patients received preoperative MR, there was a lot of patients excluded due to suboptimal image quality such as motion artifact or failure of DWI scan, we will address issue more detail in the method section.

Author action to point 1:

We revised the method section, paragraph 1, line 75-80, page 2-3

“This retrospective study was conducted by enrolling patients from the histopathology database in our institution. The preoperative diagnosis of HCC was made by the typical image appearance of the three-phase enhanced contrast-enhanced magnetic resonance imaging (MRI) including DWI study. 416 patients with complete preoperative MR study were enrolled who underwent hepatectomy or liver transplantation for curative treatment at the Department of General Surgery Kaohsiung Chang-Gung Memorial Hospital from November 2012 to March 2021.”

Line 90-91, page 3

Add 4th exclusion criteria” and 131 patients with suboptimal image quality or failure of DWI study”

Revised figure 1

2) discuss the role of LI-RADS classification as a potential and easily applicable predictor of pathological features of HCC (10.3390/diagnostics12010160)

Author Response to point 2:

 We appreciate your suggestion to bring LI-RADs to our attention to discuss the relationship of MVI and LI-RADS. After further reference articles were reviewed, we will add a new paragraph to address this issue in the discussion section.

Author action to point 2:

Add a new paragraph to address this issue in the discussion section. Line 377-397, page 12

“The liver imaging reporting and data system (LI-RADS) which was introduced by the American College of Radiology is a system of standardized terminology and a classification system for imaging findings in liver lesions. Its score classifies a liver lesion’s relative risk for HCC from LR-1 (definitely benign) to LR-5 (definitely HCC). [52] Based on the latest version of Li-RADS revised in 2018 (LI-RADS v2018), Chen et al. in 2019 [53] concluded that the categorized LR-5 lesion with mosaic architecture and non-smooth tumor margin independently predicted MVI, and Centonze et al. in 2022 [54] reported the LR-5 lesion associated with unfavorable pathological characteristics including MVI (41.7%), satellitosis (25.8), and capsule infiltration (28%), in LR 3/4 lesion, there was a non-negligible percentage of MVI (22.6%), satellitosis (9.4%), and capsule infiltration (11.3%). These results suggested that the higher LI-RADS category may predict MVI but the intermediate-risk LI-RADS category is not warranted MVI negative. Previous studies [42, 55] reported several MR image parameters to predict MVI such as tumor capsule, tumor margin, peritumoral enhancement, and peritumoral hypointensity in the hepatobiliary phase after gadoxetic acid contrast medium usage. Some of these image findings are included in the ancillary features of LI-RADS v2018.  Few studies [56,57] have discussed the ADC value in the characterization of hepatic focal lesions in correlation to LI-RADS. In future work, integration of the quantitative ADC value with the ancillary feature of LI-RADS might be helpful in improving MVI prediction between the high and intermediate-risk categories of LI-RADS.”

Add these articles in the reference list:

  1. American College of Radiology. CT/MRI Liver imaging reporting and data system v2018 core. https://www.acr.org/Clinical-Resources/Reporting-and-Data-Systems/LI-RADS/CT-MRI-LI-RADS-v2018.
  2. Chen J, Zhou J, Kuang S, Zhang Y, Xie S, He B, Deng Y, Yang H, Shan Q, Wu J, Sirlin CB, Wang J. Liver Imaging Reporting and Data System Category 5: MRI Predictors of Microvascular Invasion and Recurrence After Hepatectomy for Hepatocellular Carcinoma. AJR Am J Roentgenol. 2019 Oct;213(4):821-830. doi: 10.2214/AJR.19.21168. Epub 2019 May 23. Erratum in: AJR Am J Roentgenol. 2019 Oct;213(4):958. PMID: 31120791.
  3. Centonze L, De Carlis R, Vella I, Carbonaro L, Incarbone N, Palmieri L, Sgrazzutti C, Ficarelli A, Valsecchi MG, Dello Iacono U, Lauterio A, Bernasconi D, Vanzulli A, De Carlis L. From LI-RADS Classification to HCC Pathology: A Retrospective Single-Institution Analysis of Clinico-Pathological Features Affecting Oncological Outcomes after Curative Surgery. Diagnostics (Basel). 2022 Jan 10;12(1):160. doi: 10.3390/diagnostics12010160. PMID: 35054327; PMCID: PMC8775107.
  4. Ünal E, İdilman İS, Akata D, Özmen MN, Karçaaltıncaba M. Microvascular invasion in hepatocellular carcinoma. Diagn Interv Radiol. 2016 Mar-Apr;22(2):125-32. doi: 10.5152/dir.2015.15125. PMID: 26782155; PMCID: PMC4790063.
  5. Basha MAA, Refaat R, Mohammad FF, Khamis MEM, El-Maghraby AM, El Sammak AA, Al-Molla RM, Mohamed HAE, Alnaggar AA, Hassan HA, Azmy TM, Alaa Eldin AM, Assy MM, AlAzzazy MZ, Altaher KM, Tantawy HF, Saber S, Amin MI, Alsowey AM, Radwan MHS, Taha HF, Fathy T, Hanafy AS, Abdelbary EH. The utility of diffusion-weighted imaging in improving the sensitivity of LI-RADS classification of small hepatic observations suspected of malignancy. Abdom Radiol (NY). 2019 May;44(5):1773-1784. doi: 10.1007/s00261-018-01887-z. PMID: 30603882.
  6. Saleh GA, Razek AAKA, El-Serougy LG, Shabana W, El-Wahab RA. The value of the apparent diffusion coefficient value in the Liver Imaging Reporting and Data System (LI-RADS) version 2018. Pol J Radiol. 2022 Jan 17;87:e43-e50. doi: 10.5114/pjr.2022.113193. PMID: 35140827; PMCID: PMC8814898.

3) Include the limited number of patients as a major limitation of the study

Author Response to point 3:

Thanks for your suggestion, we will address the number of patients as the limitation of the study.

Author action to point 3:

Line 397-401, page 12

“First, although we did our best to enroll consecutive patients, there were many patients lack preoperative MR studies during the early years of the studied interval or suboptimal image quality and failure of DWI resulted in the limited number of studied patients as the major limitation of this research, besides, the nature of the retrospective study inevitably has selection bias.”

We have studied your comments carefully and have made amendments which are marked in red in the manuscript. We have tried our best to revise our manuscript in accordance with the comments.

We would like to express our gratitude to you for the comments and suggestions regarding our manuscript. It will be highly appreciated if the paper in its present revised form is accepted for publication in your esteemed journal. If there are any further comments or necessary adjustments, please do not hesitate to let us know.

Thank you and best regards.

Best regards,
Dr. Chien Chang Liao
Department of Radiology
Kaohsiung Chang-Gung Memorial Hospital, Taiwan.
+886 975 056 724
liao1009@gmail.com

Round 2

Reviewer 2 Report

The Authors provided a valuable revision of their original manuscript

The paper is suitable for publication